# A Spotlight on T Lymphocytes in Duchenne Muscular Dystrophy—Not Just a Muscle Defect

**DOI:** 10.3390/biomedicines10030535

**Published:** 2022-02-24

**Authors:** Chantal A. Coles, Ian Woodcock, Daniel G. Pellicci, Peter J. Houweling

**Affiliations:** 1Murdoch Children’s Research Institute (MCRI), Melbourne, VIC 3052, Australia; ian.woodcock@rch.org.au (I.W.); dan.pellicci@mcri.edu.au (D.G.P.); peter.houweling@mcri.edu.au (P.J.H.); 2Faculty of Veterinary and Agricultural Sciences, The University of Melbourne, VIC 3052, Australia; 3Royal Children’s Hospital, Melbourne, VIC 3052, Australia; 4Department of Paediatrics, The University of Melbourne, Melbourne, VIC 3052, Australia

**Keywords:** duchenne muscular dystrophy, T cells, inflammation, dystrophic thymus, Tregs

## Abstract

The lack of dystrophin in Duchenne muscular dystrophy (DMD) results in membrane fragility resulting in contraction-induced muscle damage and subsequent inflammation. The impact of inflammation is profound, resulting in fibrosis of skeletal muscle, the diaphragm and heart, which contributes to muscle weakness, reduced quality of life and premature death. To date, the innate immune system has been the major focus in individuals with DMD, and our understanding of the adaptive immune system, specifically T cells, is limited. Targeting the immune system has been the focus of multiple clinical trials for DMD and is considered a vital step in the development of better treatments. However, we must first have a complete picture of the involvement of the immune systems in dystrophic muscle disease to better understand how inflammation influences disease progression and severity. This review focuses on the role of T cells in DMD, highlighting the importance of looking beyond skeletal muscle when considering how the loss of dystrophin impacts disease progression. Finally, we propose that targeting T cells is a potential novel therapeutic in the treatment of DMD.

## 1. Introduction—Duchenne Muscular Dystrophy and the Immune Response

Duchenne muscular dystrophy (DMD) is a devastating, progressive X-linked neuromuscular disorder caused by mutations in the gene dystrophin (*DMD*) for which there is currently no cure. Loss of dystrophin beneath the sarcolemma (cellular membrane) of muscle fibres causes destabilization making them sensitive to damage. In response to repeated damage, inflammatory cells, including monocytes/macrophages, neutrophils, dendritic cells, natural killer cells, B lymphocytes and T lymphocytes, infiltrate skeletal muscle through continuous cycles of degeneration and regeneration [1]. Typically, the immune response linked to these repeated cycles of degeneration/regeneration contributes to the accumulation of fibrosis resulting in muscle weakness and eventually rendering the muscles non-functional. DMD patients are usually wheelchair bound by 6 to 12 years of age, and without treatment most succumb to respiratory and cardiac failure in early adult life (twenty–thirty years of age) [2,3,4].

The glucocorticoids (GC), prednisolone, prednisone and deflazacort, remain one of the few treatments for DMD. GC treatment increases ambulation by up to 3 years [3,5,6] and improves longevity in DMD patients [7,8]. However, prolonged GC use results in several adverse side effects including obesity and poor bone health, which has serious impacts on quality of life (QoL) [9]. While the mechanism of action in GC treatment is not well understood, studies have shown a reduction in inflammatory infiltrates following GC treatment in DMD [10]. Specifically, DMD patients treated with a low (0.75 mg/kg/d) and high (1.5 mg/kg/d) dose of prednisolone have shown reduced numbers of total CD2+ and CD8+ cytotoxic/suppressor T cells [10]. Furthermore, the number of myofibres invaded by T cell lymphocytes are reduced following prednisolone treatment [10], which suggests that GC treatment reduces the impact of toxic T cells in patients with DMD.

The aim of this review is to put a spotlight on T lymphocytes, highlighting their role in the pathogenesis of DMD and extending our focus beyond skeletal muscle to define the impacts of DMD on other tissues, such as thymus and spleen, which are fundamental organs for optimal immune function. Here, we describe the current understanding between the immune system and skeletal muscle pathology with a focus on the continued development of improved treatments for dystrophic muscle disease.

## 2. T Cells in Dystrophic Muscle

### 2.1. T Cells Are Active in Duchenne Muscular Dystrophy

Many studies have now shown that T cells are augmented in the peripheral blood and muscle of DMD patients [11,12,13,14]. However, some have described vast differences in T cell numbers between DMD patients. For example, CD3+ T cell counts in muscle biopsies from three patients varied greatly: 0.25, 46.03 and 4.18 cells/mm^2^ [11]. Interestingly CD68+ (M1 subset) and CD206+ (M2 subset) macrophage population proportions were constant across the three patients. Furthermore, activation of circulating CD4+ and/or CD8+ T cells has been observed between DMD patients with different dystrophin mutations [15].

CD4+ and CD8+ T cells express integrin-type extracellular matrix receptors VLA-4 (very late antigen-4, CD49e/CD29) and VLA-5 (CD49d/CD29) and are increased in the blood of DMD patients [12]. The VLA-4 (CD49d)- and VLA-5 (CD49e)-expressing T cells are α chain integrin dimers that bind fibronectin and enhance migration of inflammatory cells into muscle [12,13]. T cells positive for CD49d were found in muscle inflammatory infiltrates and both circulating CD4+CD49+ and CD8+CD49+ T cells were found to correlate with increased disease severity and a more rapid disease progression in a cohort of 75 DMD patients. Pinto-Mariz et al. (2015) concluded that CD49d is a predictive biomarker for disease progression and its reduction could be used as a therapeutic target to treat DMD.

### 2.2. Naturally Occurring Genetic Modifiers Linked to T Cell Function Influence DMD Progression

To date, single nucleotide polymorphisms (SNPs) in four genes, osteopontin (SPP1) [16,17,18], α-actinin-3 (ACTN3) [19], latent transforming growth factor binding protein (LTBP4) [18,20] and CD40 [21] have been identified as disease modifiers in DMD. Two of these genetic modifiers, CD40, and SPP1, are related to T cell function [17,18,20,21,22].

CD40 is a co-stimulatory factor vital for T cell activation, and its disruption is associated with earlier loss of ambulation in DMD individuals [21]. SPP1, an extracellular matrix molecule important in CD8+ T cell activation [22], was also associated with earlier loss of ambulation and reduced grip strength in steroid-treated DMD patients [23].

### 2.3. The Thymus Is Defective in the Mdx Mouse Model of DMD

The thymus is responsible for the development of T cells where lymphoid progenitors form CD4+ helper T cells and CD8+ cytotoxic T cells. The thymus not only controls the maturation of T cells but also self-tolerance through the negative selection of autoreactive T cells [24]. Quirico-Santos et al. (1995) first identified the architecture of the dystrophic thymus was altered, evident by thymic cortical atrophy, dense epithelial cell networks, the presence of cytokeratins (pair 818), increased vascularization and augmentation of intrathymic extra cellular matrix (ECM) components in mdx mouse [25]. Concomitantly, ECM deposition in the gastrocnemius muscle was also evident, with disruption of both the thymus and muscle architecture correlated with disease severity. The authors conclude a similar stimulus may be responsible for enhancing the altered immune environment observed in both skeletal muscle and thymus.

Recently, histological analyses of the thymic medullary and cortex of mdx mice revealed a disorganized architecture and no detectable compartmentalization of thymic medullary, unlike that seen in the thymus from wildtype mice [26]. The mdx medullary area branched to the cortex showing no clear boundaries. This was associated with a reduction in CD8+ medullary thymic epithelial cells (mTECs), which are responsible for the removal of auto-reactive T cells. The interaction between mTECs and T cells is required for the subsequent deletion of auto-reactive tissue-specific restricted antigens (TRAgs).

In the mdx medulla, CD3+ lymphocytes did not interact closely with TRAg-expressing cells (CK5+ mTEC) but spread out with looser contact implying surveillance and deletion of autoreactive T cells may be hindered. These authors concluded the altered architecture of thymus exacerbated dystrophic pathology in the mdx [26] (Figure 1). 

### 2.4. T Cell Reconstitution Reduces Muscle Fibrosis in Mdx Mice

Chronic inflammatory conditions are accompanied by irreversible fibrosis [27]. Studies using mouse models have shown that disrupting T cells and their cellular compartments can impact on fibrosis in dystrophic muscle. Severe combined immunodeficient (Scid) mice have a genetic immune deficiency that affects their B and T cells. Due to the lack of mature B and T lymphocytes, these mouse models are ideal for the xenoengraftment of human cells and tissue and provide a useful tool to assess the effects of the thymic immune response in disease. SCID mice are athymic and lack functional T cells due to the abnormal thymic stroma required for T cell development [28]. Morrison et al. (2000) were first to demonstrate that crossing the mdx and Scid mouse models (mdx-nu/nu) and removing the T cell-mediated immune response in dystrophic mice results in a reduction in ECM in the heart [29], diaphragm and hindlimb skeletal muscles [30]. The accumulation of ECM is a hallmark of fibrosis and key marker of pathology in DMD. Furthermore, reconstitution of normal thymic tissue into mdx-nu/nu mice to replenish their T cells resulted in levels of fibrosis similarly to immunocompetent mdx mice [29].

Complementary to this work, Farini et al. (2021) highlighted the involvement of T cells in dystrophic pathology by transplanting an mdx thymus into nu/nu mice (Tnumdx). Tnumdx were compared with a C57Bl thymus transplanted into nu/nu mice (TnuC57Bl) as controls. Features of dystrophic muscle were evident in Tnumdx mice, including increased necrotic myofibres, central nucleated myofibres (marker of regeneration), reduced tetanic force, increased ECM and fibrotic areas as well as CD4+ and CD8+ T cell infiltrate in skeletal muscles. The pro-inflammatory T helper 17 (Th17) differentiation marker (RORγt) and Th1 specific gene (T-bet) were also upregulated in muscle. Extensive muscle atrophy was apparent in Tnumdx mice in line with upregulation of MuRF-1, a known muscle atrophy marker [26].

This compelling evidence suggests a role for T cell disruption in dystrophic pathology. In a follow up study, a thymectomy was performed whereby both lobes of the mdx thymus were removed and CD4 and CD8 antibodies were used to deplete any remaining circulating T cells [31]. The combined thymectomy and antibody depletion had no effect on fibrosis but reduced expression of transforming growth factor beta 1 (TGFb1), which is a secreted protein that performs many cellular functions, including the control of cell growth, proliferation, differentiation and apoptosis, and is a key regulator of fibrosis [27,31]. No effect on fibrosis is surprizing considering the findings from Morrison et al. (2000) and Farini et al. (2021). The authors hypothesized that the attempts to remove all T cells was not achieved [31]. Nonetheless, this is interesting and highlights that complete absence of all mature T cells, as that achieved by crossing mdx with nu/nu mice, is required to reduce fibrosis in dystrophic muscle. Taken together these studies highlight the importance of the inflammatory response in the pathology of DMD and the involvement of T cell-mediated immunity in disease progression.

### 2.5. Antigen Presentation to T Cells in Dystrophic Muscle

Fundamental to adaptive immunity is the presentation of foreign antigens as peptides for T cell recognition. Major histocompatibility complex (MHC) expressed on antigen-presenting cells (APCs) present antigens to T cells mediating their activation and differentiation. There are two classes of MHC molecules on APC’s (macrophages/dendritic cells). MHC class I presents antigens to CD8+ cytotoxic T cells, whereas MHC class II present antigens to CD4+ T helper cells. An antigen is presented to the T cell receptor (TCR) and co-stimulation between CD28/B7 and CD40/CD40L activates the differentiation of T cells [32]. Expression of MHC is evident in DMD muscle, observed in myofibres invaded by CD8+ T cells, compared to muscle from healthy individuals, which had little expression of MHC [33,34].

The T cell receptor undergoes V-D-J gene rearrangements to enable diversity in antigen recognition. This gene rearrangement occurs by recombination of V-D-J (V-variable, D-diversity, J-joining) regions in the α and β chains of TCR and, in a smaller proportion of T cells, the γ and δ chains. Evidence of gene arrangements in the TCR of T cells was found in DMD patients [6]. Gussoni et al. (1994) found the Vβ2 transcript to be consistently expressed in DMD muscle biopsies. The amino acid motif RVSG was commonly found in the third complementary determining region (CDR3) of Vβ2 transcripts in the muscle of DMD patients [6]. In the mdx, T cells expressing Vβ8.1/8.2 predominated the pool of all TCR-Vβ+ expressing cells [35]. The Vβ8.1/8.2-expressing T cells were not present in the spleen, suggesting they may expand in dystrophic mdx muscle. This work highlights that invading T cells in dystrophic muscle transcribe similar receptors, from a recombinant rearrangement of V-D-J segments, indicating T cells are responding to a similar antigen.

The presence of dystrophin-specific T cells was detected in a study treating DMD patients with adeno-associated viral vectors containing a mini-dystrophin construct. Two out of six patients generated an unexpected T cell-mediated response to self-dystrophin epitopes prior to transgene treatment. Revertant myofibres expressing dystrophin were observed in muscle biopsies of the DMD patients. These revertant dystrophin-myofibres expressed functional, truncated epitopes of dystrophin recognized by auto-reactive T cells [36]. In addition, Flanigan et al. (2013) found dystrophin-specific T cell immunity was present in twenty out of seventy DMD patients treated with GC steroids (9/17 untreated, 11/53 treated steroids). Table 1 highlights two patients with different mutations in the dystrophin gene and a different T cell response to different dystrophin peptides. Patient 59 shows elevated CD4+ T cells to a dystrophin peptide within exons 59–69, whereas Patient 11 has increased CD8+ cytotoxic T cells to dystrophin peptide with exons 17–26 [15]. Whilst auto-reactivity to dystrophin is not apparent in all DMD patients, it poses problems for the mini-dystrophin transgene delivery. Understanding why this occurs in some patients and what influences the mixed pool of dystrophin epitopes is crucial to implementing such therapies in the clinic, as well as understanding T cell-mediated immunity in dystrophic muscle.

## 3. Targeting T Cells as Potential Therapeutic Candidates in DMD

### 3.1. Using an Antisense Oligonucleotide to Target a DMD Biomarker, CD49d on T Cells

As highlighted above, over expression of CD49d VLA-4 chain subunit correlates with increased disease severity and more rapid disease progression in DMD patients [13]. Pinto-Mariz et al. (2015) concluded that monoclonal antibodies for CD49d could be used to remove CD49d-positive cells in DMD. Following this work, an antisense oligonucleotide (ASO) to CD49d (ATL1102) was developed to target and reduce the expression of circulating CD49d. ATL1102 has shown promise in treating relapsing and remitting multiple sclerosis [37] and recently a phase 2 open label study was performed to assess the safety, efficacy and pharmacokinetic profile in boys with DMD [38].

Based on the work by Pinto-Mariz et al. (2015) showing the correlation of CD49d with disease severity in DMD, we hypothesise targeting CD49d with (ATL1102) would reduce inflammation, improve muscle regeneration, reduce fibrosis and improve muscle function in dystrophic muscle (Figure 2).

In mdx mice, treated from 9 weeks of age with ASO for Cd49d (ISIS 348574), in skeletal muscle we found a ~40% reduction in CD49d mRNA expression in skeletal muscle along with an improvement in muscle function measures [38]. In situ muscle function analyses show a high dose of Cd49d ASO (20 mg/kg/week) protected mdx muscles from the effects of eccentric muscle damage. In humans, ATL1102 was administered for 24 weeks in nine non-ambulant patients with DMD. This was considered to be safe and well tolerated, with no serious adverse events reported. However, there was no statistical difference in lymphocyte counts from baseline throughout the 24-week treatment. Furthermore, functional muscle strength measures remained stable throughout the trial period.

The proof-of-concept pre-clinical data in mdx mice supports a potential protective effect of an ASO to CD49d. Furthermore, the phase 2 open-label clinical trial has shown that ATL1102 has a good safety profile and is well tolerated. The stabilization of muscle function, compared with historical natural history data, supports the ongoing drug development program. However, further work is required to determine the impacts of this treatment in a larger cohort of DMD patients.

### 3.2. Regulatory T Cells (Tregs) and the Suppression of Inflammation in Dystrophic Muscle

A broader-spectrum approach could also benefit patients with DMD. Regulatory T cells (Tregs) are known to suppress key immune cells (T effector, B cells and APC’s) by controlling their activation and proliferation to maintain immune homeostasis and promote self-tolerance [39]. Tregs account for 5–10% of the CD4+ T cell population in healthy muscle and are identified by the co-expression of CD4, CD25, Forkhead box protein 3 (FOXP3) and low levels of CD127 [39]. Mutations in the gene FOXBP3 cause IPEX (Immune dysfunction, Polyendocrinopathy, Enteropathy, X-linked inheritance) syndrome in humans, which have dysregulated Tregs and severe autoimmunity [40]. Tregs are currently used to treat this autoimmune disease by suppression of pro-inflammatory mediators [41].

Since 2008, the adoptive transfer of polyclonal Tregs has been part of numerous clinical trials aimed to suppress the immune system. The first clinical trial used umbilical-derived polyclonal CD4+CD25+Tregs to immunosuppress donor cells following nonmyeloablative cord blood transplantation in patients with advanced Hematolytic cancer. The polyclonal transfer of Tregs has also been used safely in the suppressive immune treatment of type 1 diabetes (NCT02691247) and during liver (NCT02166177) and kidney (NCT02088931 and NCT02129881) transplantation. Whilst polyclonal Treg transfers show positive results, infusion of Tregs with TCR modified for recognition of a specific antigen offer more precise superlative effects on immunosuppression [42]. Antigen-specific Tregs generated for melanoma antigen tyrosine were expanded in vitro and administered to a tumour model in vivo [43]. The expanded Tregs maintained expression of FOXP3 and functional TCRs whilst successfully blocking antigen-specific effector T cell activity in a tumour cell model [43].

The immunosuppressive role of Tregs is apparent in dystrophic muscle, contributing to enhanced muscle repair [44]. Tregs are almost absent in healthy and uninjured skeletal muscle, whereas populations of Tregs are found in necrotic lesions in dystrophic mice and DMD muscle biopsies [44,45]. Treating mdx mice with a Treg-depleting anti-CD25 antibody from 3 weeks of age (until sacrifice at 6 weeks) exacerbated muscle inflammation, evident by an increased inflammatory infiltrate, elevated IFNy (pro-inflammatory cytokine), TGFb (pro-fibrotic marker) and greater numbers of damaged myofibres [44]. Similarly, blocking the P2X receptor, part of the inflammasome, resulted in a 24% decrease in CD3+ T cells and a 1.8-fold increase in Foxp3+ Tregs [46]. Low dose IL2c increased FoxP3+ Treg and the expression of anti-inflammatory cytokine Il-10 resulting in reduced muscle inflammation and serum CK (serum biomarker of damaged myofibres) suggesting reduced muscle injury [46]. Tregs are therefore important for suppression of inflammation in dystrophic muscle, making them an ideal candidate for immune-suppressive therapy [44,46].

To date, there have been no clinical trials for the use of polyclonal or antigenic specific Tregs in DMD. When considering the work of others [44,45] it seems plausible that the use of Tregs would be an ideal candidate to trial in the dystrophic setting. However, there are a few things to consider. Adoptive transfers of polyclonal Tregs can be achieved in substantial numbers and are less challenging, but their antigen specificity is broad, and this can result in more off-target effects and generalized immunosuppression [39].

Antigen-specific Tregs are a more appealing, targeted approach in DMD; however, there is very little known about the antigens that should be targeted for Tregs to suppress effector T cells or other inflammatory subsets in DMD. Further research is therefore required to determine the specific dystrophic antigen subsets that could be targeted for Tregs to recognize and suppress their activity.

The presence of revertant dystrophic peptides that induce CD4+ and CD8+ T cell immunity in DMD may be a beneficial area to further explore such therapies. Antigen-specific Tregs to target T cells activated by dystrophic peptides may help reduce pathogenic T cell numbers similar to that seen with corticosteroid treatments. It should be noted that these revertant dystrophic peptides generating T cell-mediated immune responses differed among patients [15,36]. Some patients showed no T cell-specific dystrophic peptides [15], this may also be associated with the large differences observed in T cell numbers amongst DMD patients [11]. It is therefore critical to understand what is driving T cell immunity in the dystrophic setting before such a treatment could be trialled. Specifically identifying whether it is the presence of revertant dystrophic peptides from secondary mutations downstream of the primary mutation, stimulation from other immune cells activated from sacrolemmal damage or a combination of both.

Once the role of T cells in DMD is better understood, polyclonal or antigen-specific Treg transfer could be trialled; however, it may be that only some patients would benefit from such therapies, and therefore a greater need for T cell immune profiling is required to determine the patient-specific T cell levels. A potential way forward in this space, may be to employ a ‘precision medicine’ approach for an individual. This could account for the inconsistences in T cell numbers which may be the result of differing dystrophic peptides mediating T cell immunity. For example, a patient with low levels of T cells would not benefit from such treatments, whereas those with high levels of circulating T cells could benefit from Treg therapy. Furthermore, identifying the Tregs subsets involved in an individual DMD patient would identify the best candidates to expand and transfer. Such an approach could help avoid off target and systemic immunosuppression in the case of polyclonal Treg transfer. Again, more research is required to understand the role of different Treg subsets in DMD and a patient-specific approach would provide the most optimal subset for expansion and transfer.

## 4. Conclusions and Future Directions

Therapeutics for the treatment of DMD are limited. Corticosteroids remain the standard of care treatment for DMD. However, long-term use has side effects. The next generation of treatments including gene read-through agent Ataluren (Translarna) is now approved for use by the European Medicines Agency but is effective only for patients with specific dystrophin mutations. Eteplirsen, an exon-skipping therapeutic, has also been granted accelerated approval by the U.S Food and Drug Administration, however, it is only amenable to confirmed mutation amenable to Exon 51 skipping. Other pathways in skeletal muscle that have been targeted in pre-clinical mouse models and show promise in treating DMD include hypertrophy/growth (PI3K/mTOR/AKT) [47] and cholesterol/metabolism [48] pathways. The anti-inflammatory drugs edasalonexent and vamorolone [49] have also showed progress in clinical trials for the treatment of DMD by inhibiting the pro-inflammatory regulator NFκβ. Whilst these therapeutic targets are promising, we have shown clear evidence for a specific and targetable role of T cells in dystrophic muscle. However, further research into the involvement of the adaptive immune response is vital if we want to unravel the processes of the immune system in dystrophic-induced inflammation. Improving our understanding of how T cells are dysregulated to contribute to fibrosis in skeletal muscle is pivotal to finding therapeutics that can dampen inflammation and improve muscle function in patients with DMD. Therapeutics targeting T cells have shown benefits in other chronic inflammatory diseases which could also be used to benefit patients with DMD in the future. Combinatorial drug therapy of exon-skipping drugs targeting growth/metabolic pathways with anti-inflammatory agents targeting the innate and/or adaptive immune system could also prove to be instrumental for treatment of DMD.

## Figures and Tables

**Figure 1 biomedicines-10-00535-f001:**
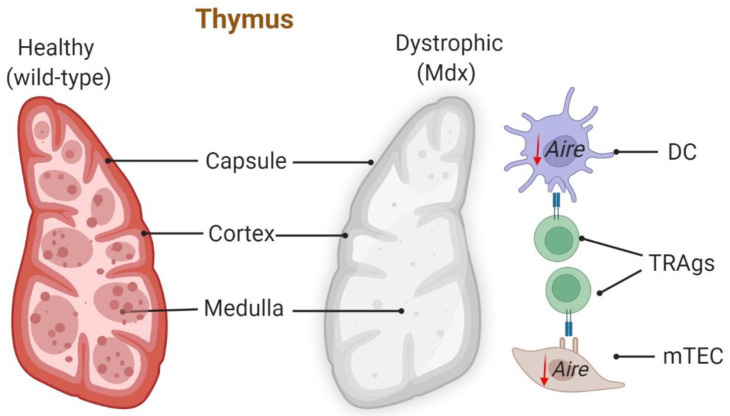
Architecture of thymus is altered in mdx. Wild-type thymus shows well-organized lobules with clear cortex and medullary boundaries. Screening and deletion of potential auto-aggressive lymphocytes (expressing tissue-restricted antigens (TRAgs)) by dendritic cells (DCs) and medullary thymic epithelial cells (mTECs) is under the control of autoimmune regulator (AIRE). In the mdx thymus, the architecture of the thymus is altered, evident by disorganized lobules with unclear cortex and medullary boundaries. The mdx thymus also has reduced medullary thymic epithelial cells (mTECs) and reduced expression of autoimmune regulator (AIRE) [26].

**Figure 2 biomedicines-10-00535-f002:**
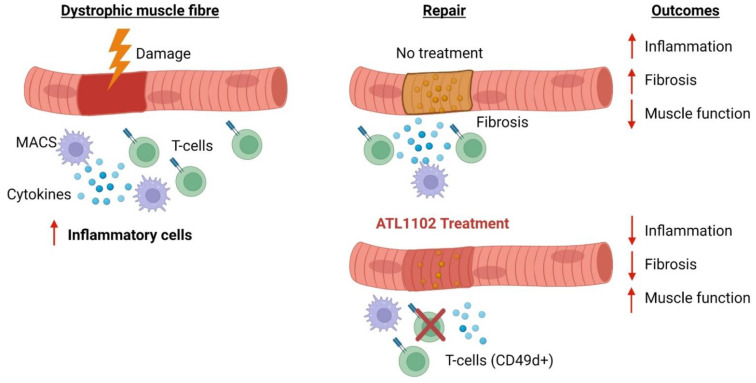
Using anti-sense oligos (ASO) to target T cells in dystrophic muscle. Repeated cycles of muscle damage in dystrophic muscle results in an influx of inflammatory cells and pro-inflammatory cytokines (MACS = macrophage, pro-inflammatory cytokines, and T-cells) leading to fibrosis and reduced muscle function. CD49d is a biomarker in DMD, with high levels of CD49d+ T-cells linked to poorer disease prognosis and earlier loss of ambulation [13]. ATL1102 is an anti-sense oligo (ASO) that has been developed to selectively target the RNA of human CD49d blocking its translation to dampen T cell response in dystrophic inflammation. We hypothesise reducing expression CD49d would result in dampening of inflammation to improve muscle regeneration, reduce fibrosis and improve muscle function.

**Table 1 biomedicines-10-00535-t001:** T cells targeting dystrophin epitopes in two DMD patients with different dystrophin mutation [15].

	Patient 59	Patient 11
**DMD mutation type**	non-sense mutation	out of frame deletion
**DMD mutation exon**	exon 69	exons 49–54
**Steroids**	prednisone	steroid naïve
**T cell response**	CD4+ T cells	CD8+ T cells cytotoxic
**Peptides upstream/downstream mutation**	upstream	upstream
**Location of antigenic dystrophin exon/peptide**	exon 60–61	AA 721–740
**Domain of dystrophin antigenic peptide**	central rod sub-domain repeat 24	central rod domain repeat 4
**Antigenic peptide sequence**	NTRWKLLQVAVEDRVRQLHE	n/a
**Dystrophin antigenic peptide pool**	DP8 exons 59–69	DP3 exons 17–26

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
