# Peer review of "A Spotlight on T Lymphocytes in Duchenne Muscular Dystrophy—Not Just a Muscle Defect"

_biomedicines, 2022, doi:10.3390/biomedicines10030535_

Round 1

Reviewer 1 Report

The authors present a comprehensive review of the role of T lymphocytes in the pathogenesis of Duchenne muscular dystrophy, including  the main articles published on the subject. Please correct or clarify the following points:

1. line 13 (Abstract) "...fibrosis of skeletal muscle, diaphragm and heart, which causes muscle weakness", because fibrosis is not the only cause of muscle weakness, my suggestion is change to ... which contributes to muscle weakness...;

2. line 20 (Abstract) "...considering the how the loss of dystrophin ..." please correct to ... considering how the loss of dystrophin...; 

3. line 27 (Introduction) DMD referring to the gene, should be in italics;

4. line 28 (Introduction) "... dystrophin at the sarcolemma (cellular membrane)..." in fact dystrophin is not a sarcolemmal protein, it is situated beneath the plasma membrane (ref Muscle Nerve 24: 262–272, 2001) ;

5. line 33 (Introduction) "... accumulation of fibrosis resulting in muscle weakness..." because fibrosis is not the only cause of muscle weakness, my suggestion is change to ... accumulation of fibrosis contributing to muscle weakness;

6. line 37 (Introduction) reference  3 is not supporting the statement that "... DMD patients are usually wheelchair bound by 6 to 12 years of age, and without treatment most succumb to respiratory and cardiac failure in early adult life (twenty – thirty years of age)" because it is a study performed in a mouse model, please refer one of the studies of the natural history of DMD in humans;

7. line 38 (Introduction) regarding the statement that "prednisolone and deflazacort, remain the only approved treatment for DMD", please clarify which kind of approval you mean and add reference. Usually prednisone and prednisolone are prescribed off-label for DMD and there are other drugs approved by regulatory agencies to treat some of the mutations causing DMD. Because prednisone, prednisolone and deflazacort are in fact the main farmacological treatment for DMD, regardless of the type of mutation, my suggestion is to avoid the expression "approved treatment";

8. line 151 "... the inflammatory response in the pathology of DMD the involvement of T cell-mediated immunity, ..." please correct to ... pathology of DMD and the involvement ...;

9. line 157 the acronym TCR first appear in the text, therefore T cell receptor should be written in full;

10. table 1 - the word prednisolone is misspelled;

11. line 227 "... drug development program however, ..." please correct to ... drug development program. However, ..." 

Reviewer 2 Report

The authors summarized a unique point of view on the role of T lymphocytes in the pathomechanism of Duchenne muscular dystrophy. It is well written and comprehensive. However, I have 2 concerns.

1. line 38: It is not true that the corticosteroid is the only treatment option. Ataluren has been approved in Europe and several exon skipping agents have been approved in USA. The authors are encouraged to include how those treatment (may have) influenced the immune system.

2. Drugs targeting NF-kB (edasalonexent, vamorolone) are under clinical trial. IL6 blockade is actively under investigation. It is worth mentioning those agents.

Reviewer 3 Report

The ABSTRACT "The lack of dystrophin in Duchenne muscular dystrophy (DMD) results in membrane fragility resulting in contraction-induced muscle damage and subsequently inflammation. The impact of inflammation is profound resulting in fibrosis of skeletal muscle, diaphragm and heart, which causes muscle weakness, reduced quality of life and premature death. To date the innate immune system has been the major focus in individuals with DMD, and our understanding of the adaptive immune system, specifically T cells, is limited. Targeting the immune system has been the focus of multiple clinical trials for DMD and is considered a vital step in the development of better treatments. However, we must first have a complete picture of the involvement of the immune systems in dystrophic muscle disease to better understand how inflammation influences disease progression and severity. This review focuses on the role of T cells in DMD, highlighting the importance of looking beyond skeletal muscle when considering the how the loss of dystrophin impacts disease progression. Finally, we propose that targeting T cells is a potential novel therapeutic in the treatment of DMD." is a honest summary of a well desiged and implemented review (with some suggestions for future research).

I have not criticisms or suggestions on the typescript, but for the general readers, I strongly suggest to add and discuss a few more References to attract attention of the readers on eventual contro-indications of the Authors' proposal, but also to support the need to add to Distrophin-related therapies other molecular goals.

A few examples follow:

Vaughan D, Ritvos O, Mitchell R, Kretz O, Lalowski M, Amthor H, Chambers D, Matsakas A, Pasternack A, Collins-Hooper H, Ballesteros R, Huber TB, Denecke B, Widera D, Mukherjee A, Patel K. Inhibition of Activin/Myostatin signalling induces skeletal muscle hypertrophy but impairs mouse testicular development. Eur J Transl Myol. 2020 Apr 1;30(1):8737. doi: 10.4081/ejtm.2019.8737. eCollection 2020 Apr 7.

Vu Hong A, Sanson M, Richard I, Israeli D. A revised model for mitochondrial dysfunction in Duchenne muscular dystrophy. Eur J Transl Myol. 2021 Sep 17;31(3):10012. doi: 10.4081/ejtm.2021.10012

Amor F, Vu Hong A, Corre G, Sanson M, Suel L, Blaie S, Servais L, Voit T, Richard I, Israeli D. Cholesterol metabolism is a potential therapeutic target in Duchenne muscular dystrophy. J Cachexia Sarcopenia Muscle. 2021 Jun;12(3):677-693. doi: 10.1002/jcsm.12708. Epub 2021 May 26.

Reviewer 4 Report

The manuscript of “A spotlight on T lymphocytes in Duchenne Muscular Dystrophy – not just a muscle defect” by Chantal A. Coles and co-authors is a well-written and detailed review of the current understanding of the relationships between the T cell immune response and skeletal muscle pathology with a focus on the development of improved treatments for dystrophic muscle. The manuscript is interesting, written at a high level, easy to read and understand. The manuscript covers a large amount of literature data and makes a significant contribution to the systematization of knowledge about the impacts of DMD on other tissues, such as thymus and spleen, which are fundamental organs for immune function. The authors have cited a large number of research articles, a significant portion of which have been published over the last years. The manuscript may be accepted for publication after minor revision.

Comments:

1. Title of the section 2.4. could be improved. It may be better to say, "T cell reconstitution reduces muscle fibrosis in mdx mice”.

2. Lines 158-159: These findings need to be described in more detail. It is necessary to describe in more detail the class of MHC molecules, cell types, etc.

Round 2

Reviewer 2 Report

The authors have cleared all my concerns.